# Multimodal Learning from Egocentric Videos and Motion Data for Action Recognition

## Abstract

Action recognition from egocentric videos remains challenging due to issues like partial visibility of the user and abrupt camera movements. To address these challenges, we propose a multimodal approach combining vision data from egocentric videos with motion data from head-mounted sensors to recognize everyday office activities like typing on a keyboard, reading a document, or drinking from a mug. To evaluate our approach, we used a dataset of egocentric videos and sensor readings from 17 subjects performing these activities. Our multimodal model fuses image features extracted from videos using deep convolutional networks with motion features from eye gaze, hand tracking, and head pose sensors. The fused representation is used to train a classifier that distinguishes between 14 activities. Our approach achieves an F1 score of 84.36%, outperforming unimodal – vision-only and sensor-only – baselines by up to 33 percentage points. The results demonstrate that body tracking technology can partly compensate for the limitations of egocentric videos, enabling more accurate activity recognition performance by $1-2$ percentage points. The inclusion of eye gaze data enhances the classification accuracy for actions that entail precise eye movements, such as reading and using a phone.

**Keywords:** Multimodal Learning, Action Recognition, Deep Learning, Egocentric Videos, Eye Gaze, Hand Tracking, Head Pose, Fusion Strategy

## 1. Introduction

Egocentric action recognition (EAR) refers to the process of classifying human actions from an egocentric point of view, i.e., the person wearing the camera carries out the action Núñez-Marcos et al. (2022). EAR is a fundamental feature in robotics or AR systems to enable downstream applications, such as contextual recommendations or reminders. By analyzing the user's past data, the egocentric device can extract their preferences and offer personal recommendations for taking breaks, the type of work activity to perform, the type of snack to eat, and more.

Vision data from an egocentric point of view can be very dynamic and often unpredictable, presenting a higher level of complexity when compared to standard action recognition from fixed cameras Plizzari et al. (2023). The limited field of view of wearable devices poses a challenge for activity recognition since users are largely out of frame, resulting in partial observability of their body, often limited to hand motion. Moreover, sharp movements in the videos are caused by the natural head motion of the user, possibly producing fast changes in the image and motion blur issues Singh et al. (2016). To address these challenges, we propose using complementary sensor cues to compensate for the limitations of the visual modality. This paper focuses on leveraging hand tracking, head pose, and eye gaze data in addition to videos to infer actions in an office setting. Modern wearable devices with integrated sensors enable estimating these data by tracking the user's motion.

To this goal, we propose a multimodal learning approach for egocentric action recognition that utilizes information from not only videos but also motion data. To fuse the two data streams, we propose using a decision-level fusion strategy with element-wise multiplication, which outperforms layer-level fusion strategies by 7 percentage points. We also studied the specific contribution of gaze tracking on the final results and found that eye gaze data increases the classification performance of actions involving fine-grained eye movements such as reading and using a phone). To evaluate our approach, we use a dataset collected from 17 participants while performing 14 different actions at an office desk.

## 2. Related Work

Traditional action recognition uses third-person views from static or handheld cameras. Egocentric action recognition (EAR) has gained significant research interest in recent years, thanks to the proliferation of affordable wearables, which has enabled the collection of large behavioral datasets for analysis. Advances in wearable technology allow continuous and unobtrusive data gathering, enabling real-time action recognition. The emergence of egocentric wearables represents a paradigm shift in action recognition, transitioning the perspective from third- to first-person views Plizzari et al. (2023). Several solutions have been proposed to tackle the challenge of egocentric action recognition. Such approaches can be classified into *object-based*, *motion-based*, and *hybrid methods* Núñez-Marcos et al. (2022); Bandini and Zariffa (2020).

**Object-based methods** often rely on the prior estimation of user and object positions, and a further processing step aimed at classifying a user's action based on these. Objects are often classified as active, directly and currently used or manipulated by the user, and inactive, present in the scene but irrelevant to the current action Nguyen et al. (2016). For example, Liu et al. (2020) proposes a method based on a hand segmentation system to improve object localization, stressing the contribution of hand motion in active object detection. In another, case a primary region is identified based on the detection of the user's body, and secondary scenes are populated by localizing objects around it Gkioxari et al. (2015) Active and passive objects can also be detected depending on their appearance, based on the assumption that an object being used will look different from a passive one Pirsiavash and Ramanan (2012); Matsuo et al. (2014).

**Motion-based methods** traditionally rely on eye-, hand-, and body-tracking. While eye motion can be used in combination with object detection to create a Region Of Interest (ROI) Fathi et al. (2011, 2012), it can also be directly linked to a specific action for classification Yu and Ballard (2002). In the same way, hand motion can be used for action recognition as a standalone feature, without exploiting info from object detection Bambach et al. (2015). Cai et al. (2018) and Garcia-Hernando et al. (2018) aim at identifying actions based on hand-object interaction from hand tracking, and study the efficiency of a set of different hand features for this task. Another feature commonly utilized in EAR, ego-motion, refers to the motion generated by the user's head motion while looking at the scene from a first-person view. Ego-motion has been explored in this context both alone Kitani et al. (2011) and in combination with eye tracking Ogaki et al. (2012). Several studies have combined visual features with motion or gaze features for egocentric action recognition Spriggs et al. (2009); Shiga et al. (2014); Singh et al. (2016). For example, Shiga

et al. (2014) recognized six daily actions (watching, writing, reading, typing, chatting, walking) using egocentric images and gaze motion. They trained separate SVM classifiers for each modality and fused the probability outputs to obtain the final prediction. Similarly, Singh et al. (2016) combined visual and motion features for first-person action recognition. Nguyen et al. (2016) reviewed methods for the recognition of activities in daily life using egocentric vision. Bandini and Zariffa (2020) reviewed approaches that analyzed the hands in egocentric vision.

**Hybrid methods** tend to combine object-based and motion-based features to improve the accuracy of action recognition. A popular architecture, in this context, is the two-stream architecture Simonyan and Zisserman (2014). In this case, vision and motion features are embedded into the same space by feeding twin neural networks with RGB and Optic Flow (OF) data. It has been shown that feeding OF info into an estimator is more effective than expecting the neural network to figure it out itself. On top of the two-stream info, more channels can be added to improve estimation such as depth Tang et al. (2018) or object features Furnari and Farinella (2019). These works demonstrate that multimodal fusion of visual and non-visual sensor data can improve egocentric action recognition over unimodal approaches. The additional modalities help compensate for the limitations of egocentric videos by providing complementary information about the user's actions.

Previously, the work in Szegedy et al. (2014) has taken up the task of egocentric activity recognition in an office setting. Activities and actions represent different semantic levels as discussed by Nguyen et al. (2016). Indeed, 'actions' are atomic and do not last over longer periods of time (seconds to minutes). Further, in our work, actions only involve hand-object interactions, while most datasets contain activities where no objects are involved. For instance, Bock et al. (2023) introduced the WEAR dataset, which contains egocentric vision and inertial-based human activity recognition. WEAR contains data from 18 participants performing 18 different workout activities at 10 outdoor locations. Tadesse et al. (2021) introduce the BON dataset, which has been collected in an office setting, similar to ours. The dataset includes activities such as, e.g., *walking*, *chatting* that do not require object interactions. We build upon this work and use a dataset that includes hand-object actions.

## 3. Methods

In this section, we provide a detailed overview of the dataset used to assess the efficacy of our multimodal methodology. We further show the steps undertaken for data cleaning and preprocessing, and the different machine learning techniques experimented on during the study.

### 3.1. Dataset and Benchmark Tasks

To evaluate our approach, we used a dataset collected from 17 participants (3 females and 14 males). During the data collection, participants sat at an office desk, wore the Magic Leap 2 device[1], and performed 14 actions. The Magic Leap 2 device is an AR device that collects egocentric videos and motion data. Egocentric videos refer to video recordings from the front-facing grayscale camera. Motion data, on the other hand, consists of 3D

---

1. https://www.magicleap.com/magic-leap-2

hand(s) keypoints from the egocentric camera, eye gaze target point in 3D space, and quaternion representation for the head pose. The videos were collected at 30fps and they include recordings of the desk, the objects on it, and the hands, if available in the view. The tracking data was collected at a sampling frequency of 60fps. Participants performed 14 actions: *doing nothing*, *typing on keyboard*, *using mouse*, *using laptop*, *using phone*, *drinking from mug*, *drinking from glass*, *holding bottle*, *eating snack*, *eating yogurt*, *eating fruit*, *reading*, *taking notes*, and *using tablet*. They performed each action for 30 seconds and were asked to act as naturally as possible. The experimenter kept track of the timestamp and type of action being performed. We defined a multiclass classification task with the goal of determining the type of action performed by the subject using the video recordings, hand keypoints, eye gaze, and head pose data.

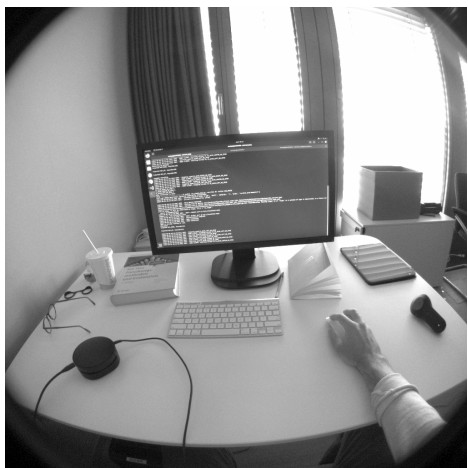 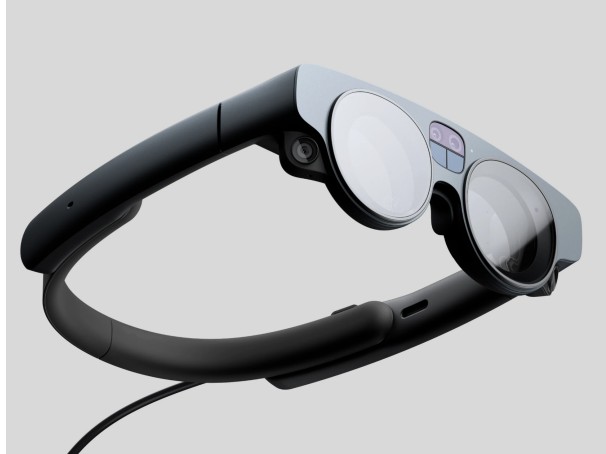

Figure 1: Example of the data collection setup during the *using mouse* action (left) and the device used to collect the data (right).

Before deciding to use this dataset, we first explored existing multimodal datasets in the literature for egocentric action recognition. Table 1 shows an overview of similar datasets. The dataset used in this work is the only one with both egocentric video recordings, as well as eye gaze, hand pose, and head pose, which allows us to evaluate our multimodal learning approach and investigate the impact of eye gaze in egocentric action recognition.

### 3.2. Data Preprocessing

**Hand-tracking data.** The hand tracking data consists of 22 keypoints for each hand, which refer to 4 joint keypoints for each finger (in total 20 keypoints), 1 keypoint for the hand wrist, and 1 keypoint for the hand center. Each keypoint is a 3D coordinate. For both hands, we have in total 44 key points or 132 features. The hand center position is computed as the mean of other keypoints. The 3D hand keypoints are represented with respect to the headset's coordinate frame, and each user has a different hand size. For these reasons,

| Dataset | Modality | Classes | Subjects | Setting |
|---------|----------|---------|----------|---------|
| BON
Tadesse et al. (2021) | RGB videos | 21 | 25 | Office |
| H2O
Kwon et al. (2021) | Object Mesh
Pose
3D point cloud | 36 | 6 | Kitchen |
| FHPA
Garcia-Hernando et al. (2018) | RGB-D videos
Hand pose
Object pose | 45 | 6 | Kitchen
Office
Social |
| HOI4D
Liu et al. (2022) | RGB-D videos | 800 | 4 | Indoor |
| ARCTIC
Fan et al. (2023) | RGB-D videos
Hand mesh
Object mesh | 11 | 10 | Indoor |
| **Our dataset** | Grayscale videos
Eye gaze
Hand pose
Head pose | 14 | 17 | Office |

Table 1: **Overview of the egocentric datasets with hands and objects involved in the field of view. The dataset we use in this work has a higher number of modalities and has been collected from a larger number of participants.**

we normalized the 3D keypoints across different subjects and experiments, to make them invariant to translation, rotation, and scale. Here we explain the three normalization steps:

- *Translation invariance* – We first center the head-relative hand keypoints with respect to the hand wrist, and set it to be the space's origin.

- *Rotation invariance* – We then rotate each hand such that: the vector connecting the hand wrist and the middle finger metacarpal joint aligns with the global vertical axis, and the vector connecting the index finger metacarpal joint and the little finger metacarpal joint aligns with the global horizontal axis.

- *Scale invariance* – Finally, we normalize the bone lengths with respect to the length of the vector connecting the hand wrist and middle finger metacarpal joint.

**Video streams.** We remove the data frames where none of the user's hands were present in the field of view using the hand tracking data to mark the presence of hands in the frame.

### 3.3. Data Synchronization

Data collection involving multiple modalities is often a challenging task. This is first because the acquisition of multiple sources of data needs to be managed simultaneously. Then the data from each input source must be consistent with other sources. To collect the data for

hand, eye gaze, and head pose tracking, we used the internal APIs of Magic Leap. Given that the sampling frequency for egocentric videos and motion data was not the same and the startup times for each module responsible for collecting each modality type were different, the collected data were misaligned. To resolve this issue we synchronize the data sources such that - for each image frame, we have a corresponding frame for the motion data at the same time instant. Since the motion data collection runs at double the frequency of the camera capture, a naive strategy would be to resample frames from the motion data such that we drop every second frame. However, due to startup time discrepancies, we will still have issues. Hence, for each frame in the image data, we pick the most recently collected frame of the motion data, i.e. temporally closest and earlier than the camera frame timestamp.

### 3.4. Data Augmentation

To increase the robustness and generalization of our approach to new, unseen data, we apply data augmentation techniques on the image frames used for training. Given an image $x$, two corresponding views $x_1$, and $x_2$ are created by applying the random transformation $t \in \mathcal{T}$, where $\mathcal{T}$ includes *random rotation* and *random flip* transformations. We applied random rotation, between $-30°$ to $30°$, perpendicular to the image plane, as it aligns with the possible amount of head tilt users have while wearing the glasses. We performed random flip augmentation around the vertical axis to make the learning process invariant to the user's handedness.

### 4. Experiments

In this section, we present our multimodal approach for egocentric action recognition using videos, eye gaze, head pose, and hand tracking data.

### 4.1. Unimodal Classification: Inception-v3 network

To process vision data, we use the *Inception-v3* network proposed by Szegedy et al. (2014) network because it has been shown to be more efficient and accurate than similar architectures. Figure 2 presents an overview of the *Inception-v3* architecture. *Inception-v3* was trained on ImageNet dataset Deng et al. (2009), which contains 1000 classes of commonly used objects. To learn the features more specific to our scenario, we re-train the *Inception-v3* network to fine-tune it to our dataset: we modify the number of output neurons in the last layer to 14, as the number of classes in our dataset.

### 4.2. Unimodal Classification: KeypointNet

To investigate the performance of motion data for action recognition, we employed a 3 layer fully-connected neural network (FCNN) model. Figure 3 shows the architecture of the model, which we refer to as *KeypointNet*. The network takes as input a 139-dimensional tensor when the three groups of data, eye gaze, hand tracking and head pose, are used. This tensor consists of 3D points for 44 keypoints of two hands, 3D point of convergence for eye gaze vectors and a 4 dimensions of the head pose quaternions. The raw data are

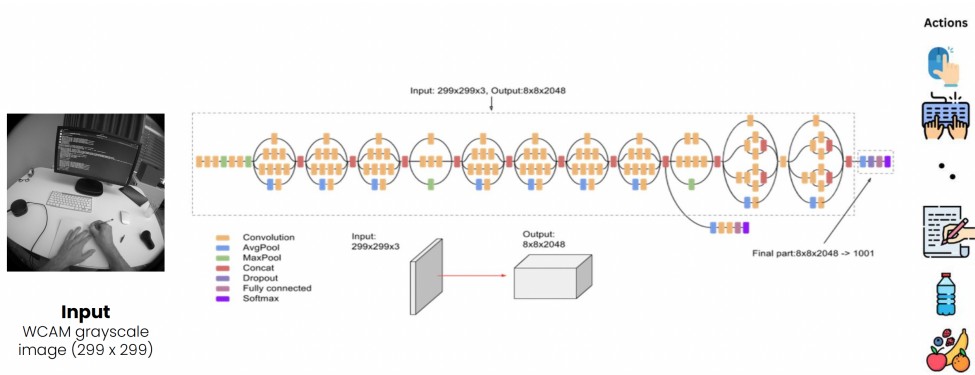

Figure 2: Inception-v3: Architecture adapted from Chang et al. (2019).

processed with a batch normalization layer and with a dropout rate of 0.8. The KeypointNet consists of three layers, each with 512, 256 and 128 neurons, respectively. These layers learn representation from raw motion data. For each layer, we use ReLU activation function. The output of KeypointNet is provided by a softmax layer. We train the model on the 14 classes of actions, which we define as a multiclass classification task.

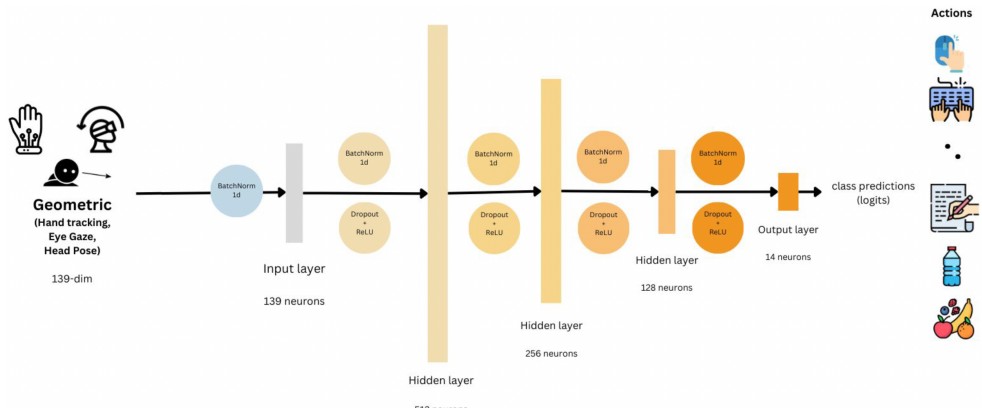

Figure 3: Overview of the architecture of KeypointNet, which takes as input eye gaze data only or in combination with 3D hand keypoints and head pose.

### 4.3. Multimodal Classification: MixNet

To fuse the two modalities, we first discard the classifiers KeypointNet and *Inception-v3* network and combine the two separate pre-trained modality-specific modules. We then optimize a single loss over the entire merged network end-to-end. Figure 4 presents an overview of the *MixNet* model architecture proposed in this work. To obtain a final classification from the two unimodal networks, we investigate two possible types of fusion strategies: *Layer-Level Fusion* and *Decision-Level Fusion* described as follows.

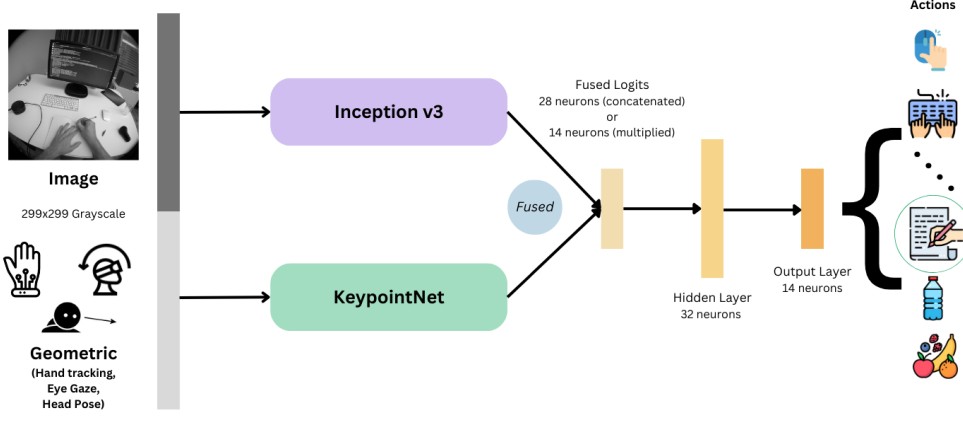

Figure 4: Architecture of MixNet that takes as input the vision (e.g., grayscale images) and motion data (e.g., hand tracking, eye gaze, and head pose). First, we pre-train the modality-specific encoders and classifiers independently. The classifier's output were then either concatenated or element-wise multiplied. The combined representations are then passed as input to a fully connected neural network.

**Layer-Level Fusion.** By design, the *Inception-v3* based classifier has an output of 2048-dimensional tensor, before it is fed to the fully connected output layer. Similarly, KeypointNet has a penultimate layer of size 128. In this experiment, we treat these tensors as the learned features by each unimodal model. Given that the latent space dimensions of the two networks are different, we downsample the 2048-dimensional feature to 128 via a trainable linear layer of neurons to keep the dimensions the same.

**Decision-Level Fusion.** Each unimodal classifier generated a 14-dimensional tensor with probabilities of the sample belonging to one of the 14 classes. We considered each output tensor as the learned feature and passed it forward for fusion. Since the probabilities for each unimodal model were in the same range (e.g., [0, 1]), we did not perform further processing on these values. To obtain a final decision, we investigated two possible approaches: *concatenation* and *multiplication*. The first group of experiments refers to the combination of probabilities obtained from each unimodal by simple concatenation. This technique has been used in prior literature, showing promising results Kwon et al. (2018); Jiang et al. (2020); Verma et al. (2018). Multiplication refers to the combination of probabilities obtained from unimodal networks by element-wise multiplication. We model KeypointNet as a multilabel, multiclass classifier because a particular body tracking configuration could be similar for multiple actions.

### 4.4. Evaluation

*Procedure.* To evaluate the performance of the multiclass classification approach, we randomly split the dataset into 90% for training and 10% for test sets using the participant identifier. We report the final results on the test set. Out of 17 subjects, we use the data

from 15 subjects for training the models, and the remaining data from 2 subjects for testing. We report the weighted F1-score, which refers to the average F1-score weighted by the number of instances for each class. We compare the performance of our approach with models using only one modality to test the advantages of using a multimodal approach for egocentric action recognition. With this baseline, we also investigated the impact of eye gaze and other data types for action recognition.

## 5. Results

In this section, we report the results of the multimodal classification approach and compare it to the unimodal approaches. We then investigate the impact of layer-level and decision-level fusion strategies on the final classification accuracy. We also discuss the impact of eye gaze features on egocentric action recognition.

### 5.1. Multimodal Learning Improves Egocentric Action Recognition

Table 2 shows the F1-score for the multimodal and unimodal networks investigated in this work. From the table, we observe that the results of both multimodal and unimodal networks are significantly higher than a random guess baseline. The model that uses eye gaze, hand tracking, and head pose features in input (KeypointNet) achieves an F1 score of 50.03%, and the one that uses vision data in input (ImageNet) achieves an F1 score of 82.83%. The highest F1 scores are obtained using the MixNet model proposed in this paper, which takes as input both data streams. In particular, the MixNet recognizes activities with an F1 score of 84.36%, which is ∼ 2% higher than the ImageNet and 33 percentage points higher than the KeypointNet. These results imply that the combination of information carried by the two input modalities are able to explain to a deeper extent a user's action. The results of KeypointNet using motion data alone are the lowest. We believe this low performance is due to highly similar data points that can have different labels. For instance, head pose and eye tracking might be very similar when typing on a keyboard or using a mouse. However, this data still contributes to the MixNet to achieve the highest performance. Figure 5 shows a few examples of the correctly predicted activities (e.g., *drink_glass*, *use_mouse*, *type_keyboard*, *use_tablet*) by MixNet in the input[2].

| Method | Model | F1-score |
|---|---|---|
| Unimodal | KeypointNet | 50.03 |
| | ImageNet | 82.83 |
| **Multimodal** | **MixNet (Ours)** | **84.36** |
| Random Guess | - | 7.14 |

Table 2: Comparison of performance of the multimodal and unimodal approaches.

---

2. Note to the reviewers, we added a demo of the performance of our approach in this link.

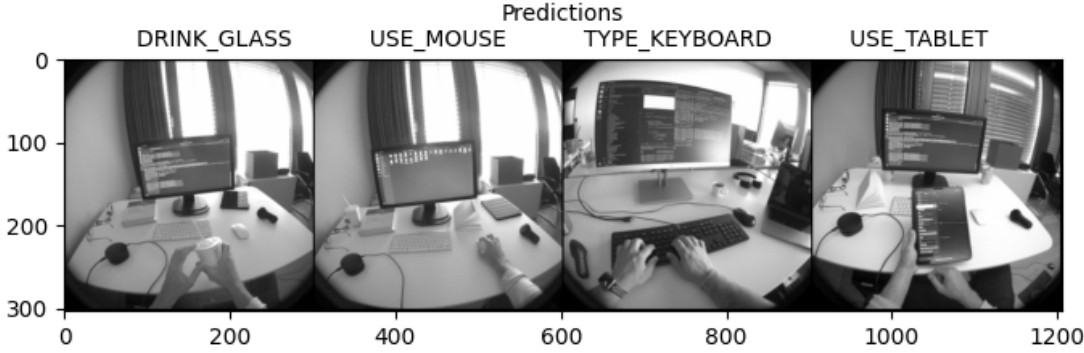

Figure 5: Example of activities classified correctly by MixNet.

## 5.2. Decision-Level Fusion Outperforms Layer-Level Fusion for Egocentric Action Recognition

We explore the impact of fusion level and fusion method on the overall task of egocentric action recognition. Table 3 presents the F1-score of MixNet using decision-level and layer-level fusion strategies for egocentric action recognition, described in Section 4.3. From the table, we observe that the highest F1 scores are obtained using the MixNet model that combines vision and motion data using the decision-level fusion strategy. The results using a decision-level fusion strategy are higher by 7 percentage points in comparison to a layer-level strategy. In particular, the F1 score of MixNet with decision-level fusion and concatenation of last layer outputs is 83.56%, and with decision-level fusion and element-wise multiplication of the last layer output is 84.36%. In contrast, the F1 scores of MixNet with layer-level fusion are 76.2% and 77.13% using concatenation or multiplication, respectively. Note that the combination of decision-level fusion and element-wise multiplication achieves the highest result so far. While the comparison between results obtained using different fusion methods is consistent for both fusion-level strategies, they differ by one percentage point.

| Fusion Level | Fusion Method | F1-score |
|---|---|---|
| Layer-Level | Concatenation | 76.2 |
| Layer-Level | Multiplication | 77.13 |
| Decision-Level | Concatenation | 83.56 |
| **Decision-Level** | **Multiplication** | **84.36** |

Table 3: Comparison of the performance for MixNet using different fusion strategies.

## 5.3. Eye Gaze Features Improve Egocentric Action Recognition

We then investigate the impact of eye gaze features on egocentric action recognition. Table 4 presents the average F1-score of the KeypointNet and MixNet using eye gaze features as

| Model | F1-score (without Gaze) | F1-score (with Gaze) |
|-------|-------------------------|----------------------|
| KeypointNet | 48.5 | **50.03** |
| MixNet | 83.33 | **84.36** |

Table 4: Comparison of the performance of KeypointNet and MixNet using eye gaze target as an input feature

.

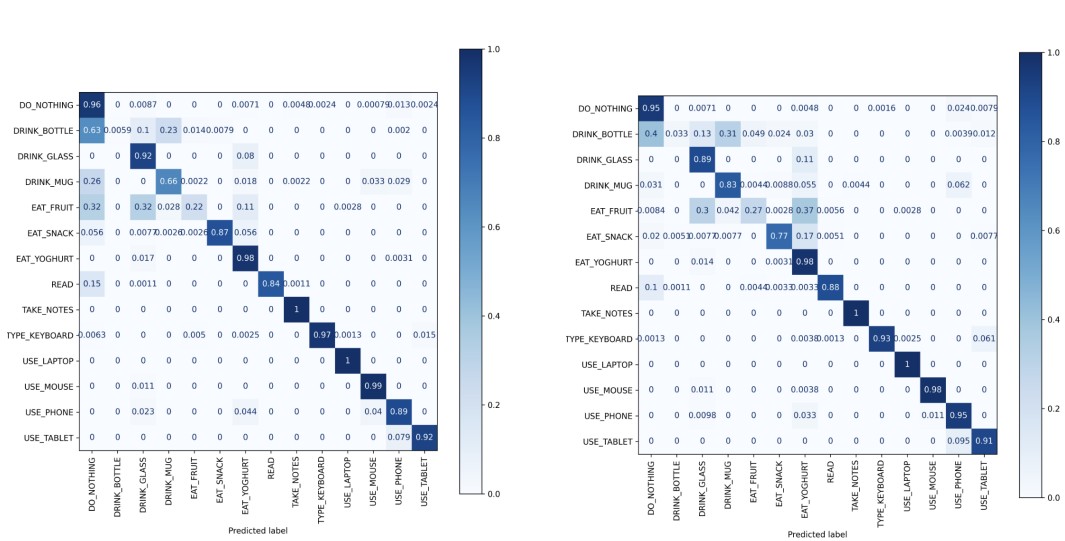

Figure 6: F1-score of the MixNet model without eye gaze as input and LOSO validation procedure.

Figure 7: F1-score for MixNet model with eye gaze as input and LOSO evaluation procedure.

input or not. In general, the eye gaze features increase the performance of KeypointNet and MixNet by 1-2 percentage points. Figures 6 and 7 present the confusion matrix of MixNet model with or without eye gaze features as input. We observe that overall the classification performance for the majority of activities remains similar hinting at the low impact of eye gaze data for such activities (e.g., do nothing, eat yoghurt). However, the MixNet increases the classification performance for the *read* action by 4 percentage points and *use_phone* activities by 6 percentage points. We believe the contribution of eye gaze features for recognizing these activities is crucial as both these activities require the user's attention and fine-grained, frequent eye movements.

## 6. Conclusions

In this work, we introduced MixNet, a multimodal deep neural network that incorporates *motion representations* – derived from eye gaze, hand tracking, and head pose – and *image*

*representations* – derived from egocentric video recording – data for action recognition. We evaluated our approach on a dataset collected from 17 participants in an office setting while they performed 14 actions – *doing nothing*, *typing on keyboard*, *using mouse*, *using laptop*, *using phone*, *drinking from mug*, *drinking from glass*, *holding bottle*, *eating snack*, *eating yogurt*, *eating fruit*, *reading*, *taking notes*, and *using tablet* – commonly performed while working. Our results show that image-based approaches can benefit from data for egocentric action recognition. Indeed, the multimodal model (MixNet) outperforms the tracking-based unimodal model (KeypointNet) by 33 percentage points, and the model that takes image data only (Inception-v3) by up to 2 percentage points. Further, the decision-level fusion strategy performs better than the layer-level strategy, hinting at the need to employ such strategies in similar benchmarks in the future. Lastly, we find that eye gaze features increase the performance of activities that require fine-grained eye movements, such as reading and using the phone, but not for other activities.

**Ethics.** MixNet is built upon a dataset that contains video recordings and motion data of 17 subjects. Participants of the study were informed about the type of data collected before volunteering to be part of the study. They signed an informed consent agreeing to participate in the experiment and to share their data for research purposes. The dataset does not contain any personally identifiable information. The video recordings captured only participants' hands.

**Limitations and Future work.** An important limitation of our work is that we evaluate our approach in a controlled setting, thus neglecting the potential data quality degradation by users' free movements (e.g., looking away from the desk). However, we asked the users to perform the actions as naturally as possible. In addition, our method excludes frames without any hands in the field of view, enabling us to pause the inference process during such occurrences. Further, the scope of actions we investigated might not be comprehensive. We believe that adding a larger and more diverse set of actions would not only further increase the capabilities of MixNet, but also cover actions performed less frequently in such scenarios. While the accuracy of MixNet was the highest, the current version of the model keeps the unimodal classifiers frozen. Experimenting with the number of trainable parameters is an interesting direction for future work. Alternatively, models such as *EfficentNet* Tan and Le (2020) and *MobileNet v2* Sandler et al. (2019), with fewer learnable parameters, hence, lower inference time can be explored. For the motion data-based classification, there are potential branches to continue the work. Firstly, the feature engineering can be redesigned to narrow down to the most statistically significant feature set for the classification task. Secondly, classifier MLP architecture can be reconsidered to include self-attention Sudhakaran and Lanz (2018) among different sources of data (hand tracking, eye gaze, and head pose). The multimodal classification task is a contemporary research challenge and hence crucial aspects such as data ingestion, feature engineering, model selection, and evaluation methods are subject to experiments and modifications. While we experimented with two different strategies for feature extraction and fusion, various methods such as self-attention, cross-modal fusion, and classifier ensemble can be explored as well. The MixNet model can be extended for additional modalities such as ambient sound and user speech to increase the performance for specific activities such as eating a fruit and drinking from a bottle.

**Implications.**    In this paper, we show that using a multimodal approach it is possible to recognize with high accuracy activities commonly performed in an office scenario from egocentric video recordings, eye gaze, hand tracking, and head pose data. Our model could be integrated into head-mounted devices for AR/VR applications in an office setting or other systems aiming to promote knowledge workers' productivity and well-being. For instance, when the model detects an action where the user interacts with an object (e.g., laptop, keyboard, phone, mug), the same object would appear in a VR setting. Another example application of the model would be to provide the user a suggestion to take a break might be generated when the model recognizes that the user has been reading for several consecutive windows Kimani et al. (2019).

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
