# OpenReview forum: "Multimodal Learning from Egocentric Videos and Motion Data for Action Recognition"
_NeurIPS.cc/2023/Workshop/Gaze_Meets_ML — Submitted to Gaze Meets ML 2023_

### Official Review · Reviewer_U5po · 2023-10-23
**Multimodal Learning from Egocentric Videos and Motion Data for Action Recognition**

**Rating:** 4
**Confidence:** 5

**Review:**

Strengths:
(+) Clear demonstration of the contribution that gaze sensor data adds in multi-modal egocentric activity classification.

Weaknesses:
(-) Evaluation: self-contained, comparing results from a hybrid approach with the two standalone inputs as opposed to broader state-of-the-art egocentric models.
(-) Evaluation: robustness, same desk environment
(-) Discussion: lacks discussion of performance from state-of-the-art models and the practical implication of the achieved scores

The paper presents a multi-modal evaluation of egocentric activity recognition. The presented results compare a model that fuses vision sensor-based recognition with sensor data from the user's hands, gaze, and head-based recognition compared with the standalone components. Layer and decision-level fusion are compared as are gaze and no gaze models. The paper is easy to read, though there are some minor grammar issues and minor details missing (example: LOSO is not defined in text, but likely stands for Leave-One-Subject-Out). The paper studies a salient task in ML and gaze. The strongest contribution of this paper is an understanding of how much improvement multi-modal models provide compared to each single modality, and the classes in which gaze improves performance the most.

However, the weaknesses in the evaluation limit the paper's contribution to the field. First, the evaluation is self-contained in that other state-of-the-art models for this task are not applied. The Hybrid model is only compared with its own vision and keypoint-based models. Second, the authors collect their own dataset consisting of 30 seconds per 14 classes times 17 users. The volume here is acceptable for small-scale evaluations but limits the scope of the claims. Evaluation of state-of-the-art or comparison on a common dataset as the literature would allow the findings to be grounded in the existing work in this space, rather than a self-contained finding that is useful (gaze helps certainly classes), but not generalizable.

This line of research is a promising direction, and I would recommend that the authors leverage their approach and compare it against state-of-the-art models on their dataset or find ways to include existing models that are not the individual components of their hybrid models. It may be the case that the state-of-the-art models only use a subset of the features/modalities available in the collected dataset when compared to MixNet. In its current state, it is not clear where the presented results stand in the broader existing work, and the paper lacks discussion of where these specific performance scores fall relative to past results or expectations on this dataset. In this case, the desk environment was uniform across all data, which is not likely to generalize. The authors could consider applying state-of-the-art models to their environment, and demonstration of better or similar performance would provide a cue that leads to further model and detection innovation. Last, while scenarios that these types of predictions are discussed near the end of the paper, the implications of the achieved performance rates or those common in literature are not detailed.

Recommendations:
-Justification of the selected classes, i.e., why does drinking from a mug matter, and justification for the length of data collection for each class (30 seconds per subject per class is quite limited)
-Include GTEA in Table 1 (https://ai.stanford.edu/~alireza/publication/ECCV12.pdf). The direct link within this paper to GTEA seems to be down, but it could be included as a row with RGB + Gaze modalities.
-Page 2, grammar "In another, case"
-Table 1, including information on the volume of collected data (X seconds per subject) would be informative and help benchmark the presented dataset relative to others.
-End of section 2, my interpretation is that actions are defined only with an object, but what about non-object activities? If there are no non-object activities here (besides DO_NOTHING) please state this clearly.
-End of section 2, the final sentence sounds like it is clarifying how you build upon the WEAR dataset work, but it is not clear how you extend it or why this is important for the reader as they transition to the Methods section.
-Page 8, "because a particular body tracking configuration could be similar for multiple actions." This is unclear in the KeypointNet context, does body tracking mean any body part?
-The url link to the demo requires a request of access, which would break anonymity if requested by the reviewer and was not viewed as part of this review as a result.
-The authors should clarify if the gaze 3D point is within the head or world coordinate frame. I.e., if they user stares at an object in front of them while rotating their head and maintaining gaze on object, would the 3D position change or be the same?
-The authors should justify why a weighted F1 Score was needed. If the activities are 30 seconds each, why would the frequency of class labels be different, and thus why would weighted and unweighted F1 scores vary?
-LOSO is not defined in text.
-Ethics statement, could be updated to say that the environment was constant and set up by researchers, and thus did not capture the subject's personal work environment, to better justify that no PII was collected.
-Table 2 uses the optimal configuration from Table 3, and results could be presented 'in-order' in terms of optimizing the hybrid model and comparing with standalone (or other models).

---

### Official Review · Reviewer_j9Ap · 2023-10-24
**A very basic and reinvented approach for multimodal learning**

**Rating:** 4
**Confidence:** 4

**Review:**

**Overview**

The authors use a multimodal learning approach to combine information from egocentric videos and other sensors for action recognition.

The manuscript is easy to read and relatively well explained. In addition, the authors collected a new dataset from 17 subjects with multiple modalities to develop their approach. However, the methodology is overly simplistic and at many points it seems like the authors are reinventing very basic ML concepts


**General comments**

- Is the dataset and code available?

- I appreciate the detail and how well-explained everything is but it often goes too far. A 3-layer neural network does not need a name, it's not a new architecture, it's rather a very simple one. Is there a reason you chose this? You also gave new names to early fusion and late fusion, namely layer-level, and decision-level fusion, respectively. It appears you are reinventing the wheel. See also the specific comments

- The Figures look really nice but they don't seem necessary here. Maybe merge Figures 2, 3, and 4 into one figure and skip all the unnecessary details?

- How does this compare to performance from other studies in the literature?

- Can we see also see an example where the prediction was incorrect?

- Why not use your approach on more datasets?

- How many total samples did you use in the end for training/validation/test? And is there a validation set?

**Methodology**

- Why did you use Inception-v3 instead of a more modern architecture, something based on ViTs?

- Where did the 139 dimensions of the input come from?

- You say 'unimodal classification' and '3 groups of data' but they seem like 3 modalities to me.

- If you are evaluating on 2 subjects, you should at least do cross-validation.


**Specific comments**

- “Our approach achieves an F1 score of 84.36%, outperforming unimodal – vision- only and sensor-only – baselines by up to 33 percentage points. The results demonstrate that body tracking technology can partly compensate for the limitations of egocentric videos, enabling more accurate activity recognition performance by 1 – 2 percentage points.”: 33 or 1-2? I know what you mean but it's confusing early on
- “Previously, the work in Szegedy et al. (2014) has taken up the task of egocentric activity recognition in an office setting. Activities and actions represent different semantic levels as discussed by Nguyen et al. (2016). Indeed, ’actions’ are atomic and do not last over longer periods of time (seconds to minutes). Further, in our work, actions only involve hand- object interactions, while most datasets contain activities where no objects are involved. For instance, Bock et al. (2023) introduced the WEAR dataset, which contains egocentric vision and inertial-based human activity recognition. WEAR contains data from 18 participants performing 18 different workout activities at 10 outdoor locations. Tadesse et al. (2021) introduce the BON dataset, which has been collected in an office setting, similar to ours. The dataset includes activities such as, e.g., walking, chatting that do not require object interactions. We build upon this work and use a dataset that includes hand-object actions.”: Can you write it in a more organized way and specifically point your contribution? I think the content is there but it's hard to follow.
- “we retrain the Inception- v3 network to fine-tune it to our dataset: we modify the number of output neurons in the last layer to 14, as the number of classes in our dataset.": do you mean you fine-tune? I'm not sure why you say retrain, it seems like you are fine-tuning
- “To learn the features more specific to our scenario”, “we modify the number of output neurons in the last layer to 14, as the number of classes in our dataset.” - unnecessary to mention, just say you fine-tuned the model and it will be clear what you did
- You do not need to mention hyper parameter values in the text. Can you add them in a supplement?
- “To fuse the two modalities, we first discard the classifiers KeypointNet and Inception-v3 network and combine the two separate pre-trained modality-specific modules” - reading this does not make sense to me but I think you just discard the classification head and use the encoder part of each model. Write it in a way that is understood
- Table 2, ImageNet is not a model
- Table 4, why aren't you showing Inception+Gaze?
- Figures 6 and 7 should be in the supplement, at most. If you want to show F1 per class that's fine or have certain observation for certain classes and you want to mention it in the text but these figures are unnecessary
- “Firstly, the feature engineering can be redesigned to narrow down to the most statistically significant feature set for the classification task.”: not sure what you mean by that

---

### Official Review · Reviewer_xEwb · 2023-10-24
**Very well written paper with limited novelty**

**Rating:** 6
**Confidence:** 5

**Review:**

The authors highlight challenges in action recognition from ego-centric videos due to partial visibility and abrupt camera movements. They propose a multi-modal approach combining vision data with motion data from sensors attached to the head of the subjects.
Overall, a clearly structured and well written paper, discussing related work and various egocentric datasets and how the dataset used by the authors compare to them. The authors do not discuss specifics of the data collection and do not provide any additional insights into the distribution. Moreover, the size limitations of the dataset suggests a lack of variety in the dataset.
The approach suggested by the authors have limited novelty and is a straight-forward combination of multiple modalities through specific encoders. The authors perform rich ablations to benchmark the impact of additional modalities, however, that’s the extent of novelty provided by the paper.
The contribution would have been much stronger if the authors provided experiments on a few of the publicly released datasets allowing us to compare with other existing approaches.
Overall, a very well written paper but with limited novelty apart from the insights into the impact of various modalities on the task of ego-centric HAR.

---

### Meta-Review · Area_Chair_9ZSJ · 2023-10-26

**Recommendation:** Reject
**Confidence:** 5

**Metareview:**

The paper presents a methodology for egocentric activity recognition using multi-modal data. As the reviewers pointed out, the collected  multimodal dataset is very important contribution. Moreover if the dataset released can have a bigger impact to the community. However as the reviewers pointed out the paper lacks any methodology novelty as well as comparative study with SOTA algorithms.

---

### Decision · Program_Chairs · 2023-10-26

Reject